# Leisure engagement in older age is related to objective and subjective experiences of aging

Jessica K. Bone [1] ✉, Feifei Bu[1], Jill K. Sonke[2] & Daisy Fancourt [1]

Leisure engagement has potential to slow health and functional decline in older age. However, the benefits of different leisure domains for different aspects of aging remains unclear. In 8771 older adults from the Health and Retirement Study (a longitudinal panel study), we measured engagement in physical, creative, cognitive, and community activities. Outcome-wide analyses used 23 aging experiences across seven domains eight years later (daily functioning, physical fitness, long-term physical health problems, heart health, weight, sleep, subjective perceptions of health). Physical activity was related to more positive experiences in all domains but heart health eight years later. Creative engagement was positively related to aging experiences in four domains longitudinally. Cognitive and community engagement were less consistently related to aging experiences. Physical and creative activities may influence important aging metrics, reducing age-related decline and keeping older adults functionally independent for longer, potentially limiting increasing healthcare costs.

Leisure engagement has been shown to be evolutionarily beneficial. Defined as voluntary non-work activities that are participated in for pleasure[1], leisure includes everyday activities such as hobbies, arts, culture, sports, educational classes, volunteering, and community groups. There is extensive evidence linking leisure engagement to physical and mental health across the life course, from the benefits of play for children's development[2] to the role of community engagement in preventing cognitive decline and dementia[3,4]. Although leisure engagement is a ubiquitous human experience, it is often undervalued. Leisure activities are often viewed as secondary to work, only engaged in when people have available time and resources. Leisure has also been commodified for capitalism[5], leading to increases in cost, limiting accessibility, and contributing to some forms of leisure becoming elite activities, only participated in by those of higher socioeconomic position[6,7].

Internationally, many aging cohorts include questions on leisure engagement, but they have not been widely used to understand aging. It thus remains unclear which leisure activities impact different aspects of aging. Theoretically, leisure engagement may help to prevent age-related decline through wide-ranging causal mechanisms, including psychological (e.g., enhancing self-efficacy), biological (e.g., reducing

levels of stress hormones), social (e.g., increasing social support), and behavioral (e.g., enhancing motivation to engage in other health behaviors) pathways[8]. Different leisure activities can be grouped and considered to affect health based on their active ingredients and the mechanisms of action that they stimulate[8], such as the extent to which they include social contact, sedentary behavior, or working towards a goal. However, previous studies have often focused on single domains of leisure, such as arts engagement[9] or physical activity[10], or have selected specific health outcomes, including depression[11], dementia[12], and wellbeing[13]. Although some studies have explored leisure engagement and broader experiences of aging, these have been relatively small and did not include nationally representative samples[14–18]. The extent to which different types of leisure activities independently influence aging experiences, including objective and subjective measures of health, requires further investigation.

The potential benefits of leisure activities become increasingly relevant for older adults. Later in life, close social ties may be lost; people are more likely to live alone, many have low income, and progressive age-related chronic diseases may limit activities[19]. As a result, maintaining positive behaviors that keep people healthy may become more challenging. The transition from work to retirement can lead to a

[1]Research Department of Behavioural Science and Health, Institute of Epidemiology & Health Care, University College London, London, UK. [2]Center for Arts in Medicine, University of Florida, Gainesville, FL, USA. ✉e-mail: jessica.bone@ucl.ac.uk

loss of activity, social contacts, and purpose[20,21] and cognitive decline[22]. However, it can also result in increased time for leisure activities. There is even some evidence that leisure activities become increasingly beneficial after retirement[23,24]. Understanding the wide-ranging potential benefits of leisure activities, beyond addressing individual deficits, is thus particularly important in older adults. It also has implications for healthcare systems internationally, given the increasing interest in referring older adults to community-based activities through social prescribing[9,25].

In this study, we take an outcome-wide approach[26,27], assessing experiences of aging related to physical health. We include 23 outcomes across seven domains: daily functioning, physical fitness, long-term physical health problems, heart health, weight, sleep, and subjective perceptions of health. Using data from the Health and Retirement Study (HRS), we compare four domains of leisure activities: physical, creative, cognitive, and community activities. We show positive associations between some domains of leisure engagement, particularly physical and creative activities, and many (but not all) age-related processes. As relationships are independent of a range of demographic, socioeconomic, and neighborhood factors, our findings suggest physical and creative activities could influence these important aging metrics. Older adults should be supported and encouraged to incorporate physical and creative activities into their everyday lives. Our findings thus support previous evidence that leisure engagement is evolutionarily beneficial, relevant to humans and their health, and should be used to understand experiences of aging.

## Results

We included 8771 older adults (55% female) who participated in HRS between 2006 and 2018. Ages ranged from 50 to 94 (mean = 63.18, standard deviation [SD] = 8.45). Overall, 85% were of White race/ethnicity, 10% Black/African American, and 5% identified as Other race/ethnicities, 71% were married, and 48% were retired (Table S3). The HRS Social Engagement questionnaire measured participation in 15 leisure activities, which have previously been categorized into physical, creative, cognitive, and community activities[28]. The frequency of engagement in each domain ranged from 0 (never) to 6 (daily). Physical activities were most frequently engaged in (mean = 3.43, SD = 1.76), followed by cognitive (mean = 2.86, SD = 1.16), creative (mean = 2.59, SD = 1.16), and community activities (mean = 1.26, SD = 0.86). Correlations between leisure domains ranged from $r = 0.21$ to $r = 0.34$.

Using an outcome-wide approach, the longitudinal associations between each activity domain and experiences of aging eight years later are presented in Fig. 1. We assessed 23 outcomes across seven domains: daily functioning, physical fitness, long-term physical health problems, heart health, weight, sleep, subjective perceptions of health (Table S4). The type of regression model and the included sample size were determined by each outcome (sample size varied because of subsample eligibility; Table S1). Models included all four leisure domains and are presented before and after adjustment for demographic, socioeconomic, and neighborhood covariates (age, gender, race/ethnicity, marital status, education, employment, pension status, household income, assets, household size, and neighborhood safety,

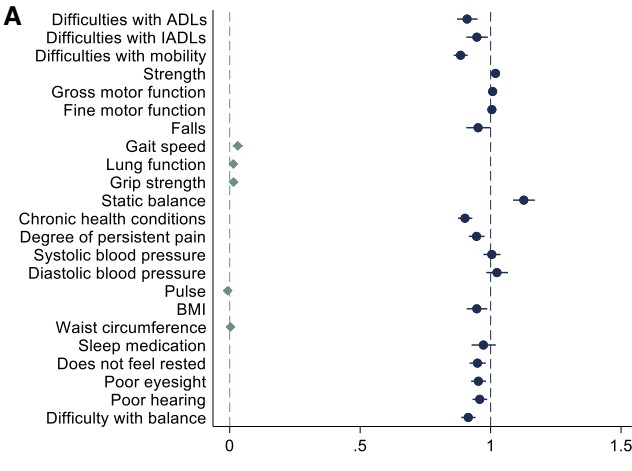

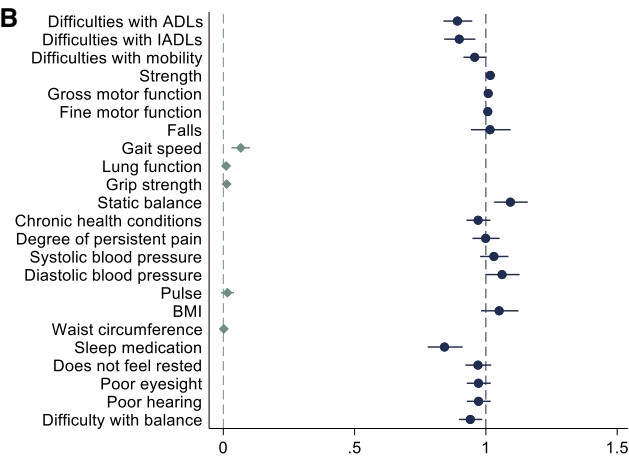

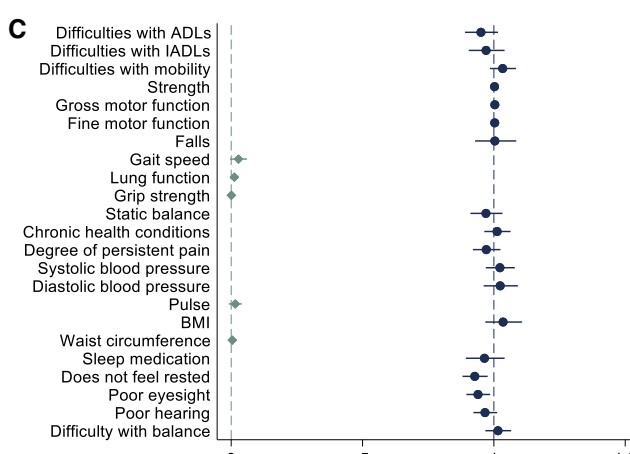

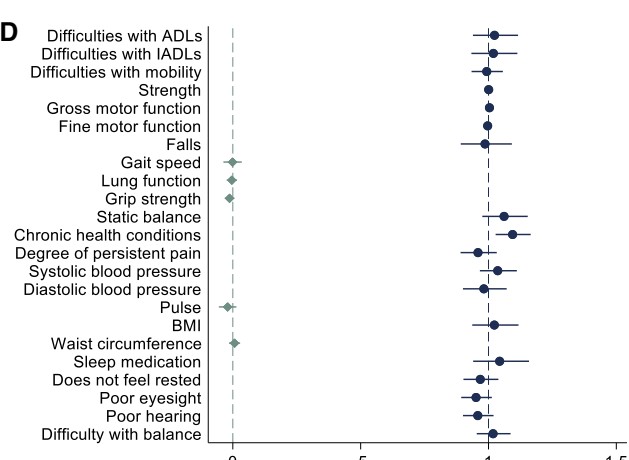

**Fig. 1 | Adjusted beta coefficients (diamonds), odds ratios (circles), and incidence rate ratios (circles) from regression models testing the longitudinal associations between frequency of engagement in the four domains of leisure activities and experiences of aging eight years later.** Data are presented as coefficients and accompanying 95% confidence intervals. Results adjusted for demographic, socioeconomic, and neighborhood covariates, weighted, and based on 20 imputed datasets. **A** Physical activities. **B** Creative activities. **C** Cognitive activities. **D** Community activities. Source data are provided as a Source Data file.

physical disorder, and social cohesion) and the baseline measure of the outcome in Tables S5 and S6. Findings from the fully adjusted models are reported here.

### Physical activities
More frequent engagement in physical activities (e.g., sport/exercise, walking) was associated with all aspects of daily functioning, physical fitness, long-term physical health problems, and subjective perceptions of health eight years later (Fig. 1). The strongest associations were for higher odds of better static balance (odds ratio [OR] = 1.13, 95% confidence interval [CI] = 1.09–1.17) and less perceived difficulties with balance (OR = 0.91, 95% CI = 0.89-0.94), and lower odds of difficulties with mobility (OR = 0.88, 95% CI = 0.86-0.91), difficulties with activities of daily living (ADLs; incidence rate ratio [IRR] = 0.91, 95% CI = 0.87–0.95), and chronic health conditions (OR = 0.90, 95% CI = 0.88–0.93). There was mixed evidence for associations between weight and sleep. For example, more frequent physical activity was associated with lower odds of being overweight but not waist circumference.

### Creative activities
More frequent engagement in creative activities (e.g., gardening, needlework, and hobbies) was associated with some aspects of daily functioning, physical fitness, sleep, and subjective perceptions of health eight years later. The strongest associations were fewer difficulties with ADLs (IRR = 0.89, 95% CI = 0.84–0.95) and instrumental activities of daily living (IADLs; OR = 0.90, 95% CI = 0.84−0.96), higher odds of good static balance (OR = 1.09, 95% CI = 1.03–1.16), and lower odds of using sleep medication (OR = 0.84, 95% CI = 0.78−0.91). However, there was no longitudinal evidence that creative engagement was associated with subsequent long-term physical health problems, heart health, or weight.

### Cognitive activities
There was only evidence for longitudinal associations between engaging in cognitive activities (e.g., reading, writing, and games) more frequently and lower odds of not feeling rested after sleep (OR = 0.93, 95% CI = 0.88−0.98) and rating eyesight as poor (OR = 0.94, 95% CI = 0.90−0.99).

### Community activities
There was no evidence for protective associations between more frequent engagement in community activities (e.g., volunteering, educational courses, and sports/social clubs) and aging experiences eight years later. However, more frequent community engagement was associated with higher odds of chronic health conditions eight years later (OR = 1.09, 95% CI = 1.03−1.17).

### Concurrent associations
Cross-sectional associations between each activity domain and experiences of aging are also presented in the Supplementary Materials (Tables S7 and S8). Concurrently, there was strong evidence that physical activities were positively associated with nearly all aspects of aging. Creative activities were also positively associated with most aspects of all domains of aging. Cognitive activities were only positively associated with subjective perceptions of eyesight but were also associated with worse outcomes in diastolic blood pressure, BMI, and waist circumference. Community activities were associated with lower odds of high systolic blood pressure, not feeling rested after sleep, rating eyesight as poor, and perceived difficulties with balance.

### Sensitivity analyses
We computed E-values as indicators of how robust findings were to potential unmeasured confounding[29]. Longitudinally, where there was evidence of associations between leisure engagement and subsequent aging experiences, E-values varied between 1.08 and 1.67 (Table S6). This indicates that any unmeasured confounders that were associated with both the exposure and outcome by risk ratios of between 1.08- and 1.67-fold, conditional on the measured demographic, socioeconomic, and neighborhood confounders, could shift the observed associations to the null.

In further sensitivity analyses, we additionally adjusted for health covariates (cognition, depressive symptoms, prescription medication, psychiatric problems, self-rated health measured at baseline) and health behavior covariates (alcohol use and smoking measured at the wave prior to baseline). After doing so, associations between physical and creative activities, daily functioning, and physical fitness remained similar (Table S9). However, associations with other aspects of aging were attenuated. There remained very little evidence for associations between cognitive and community activities and experiences of aging.

As there were concerns about potential bias due to controlling for the outcome at baseline, we also repeated the longitudinal adjusted analyses after omitting the baseline outcome measure (Table S10). Evidence for associations between leisure activities and experiences of aging remained similar, albeit slightly stronger, for physical, creative, and cognitive activities.

We also performed sensitivity analyses, including different levels of leisure engagement, to provide a more comprehensive picture of the associations with experiences of aging (Tables S11A and S11B). We categorized engagement frequencies into none, monthly, and weekly engagement. This indicated that, for physical activities, most of the associations with experiences of aging were driven by more frequent weekly engagement. There were fewer associations for participants who only engaged monthly. In contrast, for creative activities, findings were more similar across monthly and weekly engagement frequencies, showing less of a dose-response relationship. This suggests that any level of engagement in creative activities could potentially be beneficial.

Finally, due to concerns around reverse causation, we limited the sample to participants without chronic health conditions at baseline (Table S12). This resulted in small sample sizes (n = 395 to n = 1460), meaning there was very little evidence for associations between leisure engagement and experiences of aging after adjusting for the wide range of covariates. However, coefficients and confidence intervals were generally in line with the main analyses.

## Discussion
In this nationally representative sample of older adults in the US, physical activity was most consistently associated with experiences of aging. It was related to more positive aging experiences in all domains except heart health 8 years later. Engagement in creative activities was also positively related to some aspects of daily functioning, physical fitness, sleep, and subjective perceptions of health longitudinally. However, cognitive and community activity engagement were less consistently related to aging experiences. Looking across health domains, daily functioning, physical fitness, and subjective perceptions of health were most often associated with leisure engagement. Associations were independent of other domains of leisure, demographic, socioeconomic, and neighborhood factors, and previous experiences of aging. Sensitivity analyses also indicated that findings were relatively robust to unmeasured confounding and additional adjustment for health and health behavior covariates.

Given the wealth of previous evidence for the health benefits of physical activity[10,11,30–32], it is unsurprising that this form of leisure was most consistently associated with experiences of aging. The consistent associations between physical activity (sport/exercise, walking for 20 min or more) and a wide range of outcomes eight years later suggest that these activities contain a range of active ingredients that support older adults' health. Our findings extend previous evidence by showing the broad positive impacts of physical activity across

domains, allowing for comparison of standardized effects across health outcomes, and demonstrating that these associations are independent of participation in creative, community, and cognitive activities. Sensitivity analyses indicate the importance of regular physical activity, with most associations only present for weekly or more frequent engagement. Physical activity may influence experiences of aging through unique pathways not activated by other leisure activities, such as adaptations of physiological systems, including the neuromuscular system that coordinates movements, and metabolic processes regulating glucose and fatty acid metabolism[33].

As outcomes typically worsen across all domains in older age, physical activity could be used to optimize health by slowing decline and better-maintaining experiences of aging. Regardless of their engagement in other types of leisure, all older adults should be supported and encouraged to incorporate some form of exercise into their everyday lives. However, although more frequent physical activity was associated with lower BMI 8 years later, it was not associated with blood pressure, pulse, or waist circumference. This lack of longitudinal associations was unexpected in comparison to previous findings[32]. It is possible that including low-intensity exercise, such as walking, in our measure of physical activity masked longitudinal associations between higher-intensity activity and these outcomes. Future research should explore different types of physical activity separately.

The evidence for associations between creative activities (e.g., hobbies, gardening, and needlework) and experiences of aging was less consistent but generally positive. This is in line with previous inconsistent evidence. Whilst some studies have shown that creative activities provide additional benefits for older adults over other leisure activities[8,34,35], others have found less clear evidence for associations between creative activities, healthy aging[18], and cognitive decline[36]. In this study, the strongest relationships for creative activities were with daily functioning and physical fitness. By accounting for previous levels of functioning and fitness, we effectively estimated the change in these outcomes over 8 years. Doing creative activities as infrequently as monthly may help to prevent age-related decline in physical functioning and fitness, even if they do not influence all aspects of aging. It is surprising that these creative activities were more strongly associated with aging experiences than community activities (e.g., volunteering, educational courses, clubs, and organizations), which are more likely to include social interaction. This suggests that it is not just the social element of activities that can benefit older adults. Other mechanisms through which creative activities might prevent decline include maintaining cognitive flexibility[8], reducing stress and inflammation[37], and providing a sense of purpose and meaning in life[38].

The lack of consistent associations between community activities, cognitive activities (e.g., reading, writing, and playing games), and experiences of aging was unexpected. There is extensive evidence that community and cognitive activities can benefit a range of health outcomes in later life, including self-rated health, frailty, disability, chronic pain, dementia, depression, and wellbeing[4,38–44]. It is possible that community and cognitive activities are more strongly associated with mental than physical health. However, we have previously found that attending cultural activities (a form of community engagement) is associated with healthy aging, particularly physical functioning[18]. Additionally, volunteering (another form of community engagement), has been linked to a range of health outcomes[45], although a systematic review also found stronger evidence for the benefits of volunteering on mental than physical health[46]. The lack of evidence for associations with experience of aging in this study could also be due to prodromal effects, whereby people already experiencing health deficits were less likely to engage[47]. Engagement in the preceding decades would need to be measured to explore this further. Given that there were more concurrent than longitudinal associations between cognitive and community engagement and aging experiences, it is also possible that

these types of activities are perishable commodities requiring consistent engagement to achieve potential health benefits[48]. Sustained community cultural engagement has a larger impact on older adults' well-being than short-term or repeated engagement[49]. Furthermore, community activities could enhance aging experiences by reducing sedentary behavior[8]. As we adjusted for physical activity, any mediating effects of sedentary behavior may already be accounted for, making estimates of the associations between community engagement and aging experiences more conservative.

Some domains of aging experiences (e.g., daily functioning, physical fitness, and subjective perceptions of health) may be more closely linked to leisure engagement than others (e.g., long-term physical health problems, heart health, and weight). These relationships are likely bidirectional, as older adults with better health are more able to participate in leisure activities[15,38], which then further supports subsequent aging. The paucity of associations between most leisure activities and persistent pain contrasts with previous research. In older adults in the UK, regular physical activity and cultural engagement were protective against the development of chronic pain[50]. However, the previous study explored the onset of pain in people originally free from pain over longer timescales. Methodological differences could explain why not all of our findings align with previous evidence. This demonstrates the importance of outcome-wide analyses using harmonized measures. It is also interesting that the associations between leisure engagement and subjective perceptions of health were often larger than associations with objective measures of health. Objective measures of health, with deeper roots in pathological and biological processes, may be more difficult to modify. Leisure engagement could be more likely to influence older adults' attitudes towards their health than their actual physical health. This is still useful because subjective evaluations of physical health may be more strongly related to older adults' well-being than objective measures[51].

This study has a number of strengths. We compared the associations between four different types of leisure engagement, independent of other leisure domains, and adjusted for demographic, socioeconomic, and neighborhood covariates likely to play a role in leisure engagement[6,7]. We additionally adjusted for health and health behaviors in sensitivity analyses which, alongside adjusting for the baseline measure of the outcome, should help to address reverse causality. HRS is a large nationally representative cohort of older adults, making our findings more generalizable to the US population than previous small studies of leisure and experiences of aging[14–17]. Analyses were weighted using HRS sample weights to account for non-response and the complex sample design.

However, this study also has several limitations. We measured leisure engagement at baseline and thus could not compare the effects of sustained or changing engagement on experiences of aging. As HRS only started measuring these leisure activities in 2008, this study included a relatively short follow-up of eight years. Future research should explore whether any potential benefits of engagement are maintained over longer periods as more data become available. Despite including a range of covariates, sensitivity analyses indicated that our findings were susceptible to unmeasured confounding, which could be due to factors such as urbanicity, diet, and social support. We did not adjust the main analyses for health or health behavior covariates, as it is likely that these mediate the associations between leisure engagement and experiences of aging. Although we have included these covariates in sensitivity analyses, this could have led to overly conservative estimates, and there was still evidence for similar degrees of unmeasured confounding. We recognize that gender is not a binary construct, although we had to treat it as such given the way data were collected. Further, as HRS combined a range of races/ethnicities into the Other race/ethnicity category, we were not able to investigate the influence of race/ethnic identities and associated racism and cultural caste systems. These limitations underscore the challenges of

controlling for demographics and the need for improving methods for measuring and accounting for systemic oppressions such as structural racism in research[52,53]. By exploring population-level associations, we conflated individual experiences, so future research should focus on the needs of different groups who may experience distinctive stressors and outcomes. Access to leisure, the quality of leisure time, and the availability of culturally meaningful activities also differ across groups.

We found evidence for positive associations between some domains of leisure engagement, particularly physical and creative activities, and many (but not all) age-related processes at the functional and phenotypic levels. Given that relationships were independent of a range of demographic, socioeconomic, and neighborhood factors, physical and creative activities may influence these important aging metrics. Older adults should be supported and encouraged to incorporate physical and creative activities into their everyday lives. If inequalities in leisure engagement[6,7] can be reduced, this could lead to more equitable experiences of aging among older adults[54]. Internationally, policymakers are implementing systems to promote social equity and reduce inequalities by enabling practitioners to connect older adults to leisure activities through social prescribing. Our findings demonstrate the potential benefits of these systems, as they could reduce age-related decline and help older adults stay functionally independent for longer. Our findings thus support previous evidence that leisure engagement is evolutionarily beneficial, relevant to humans and their health, and should be used to understand experiences of aging.

## Methods

### Ethical approval

This research complies with all relevant ethical regulations. All participants gave informed consent, and this study has approval from the University of Florida (IRB201901792) and the University College London Research Ethics Committee (project 18839/001). HRS offers financial payments as tokens of appreciation to respondents for participating, but these were not intended as compensation.

### Sample

Participants were drawn from the Health and Retirement Study (HRS), a nationally representative study of more than 37,000 individuals over the age of 50 in the US[55]. The Health and Retirement Study (HRS) was initiated by the National Institute on Aging and conducted by the Institute for Social Research at the University of Michigan to track the Baby Boom generation's transition from work to retirement. The initial HRS cohort was interviewed for the first time in 1992 and followed up every two years, with other studies and younger cohorts merged with the initial sample. Together, these studies create a fully representative sample of individuals over the age of 50 in the United States. Further details on study design are reported elsewhere[55]. In this study, we combined seven HRS public datasets (HRS 2018 Tracker Final Release [V1.0], RAND HRS Longitudinal File 2018 [V1]; RAND HRS Detailed Imputations File 2018 [V1]; 2008 RAND HRS Fat File [V3A]; 2010 RAND HRS Fat File [V5F]; 2016 RAND HRS Fat File [V2B]; 2018 RAND HRS Fat File [V2A]). Raw data are available from HRS (https://hrsdata.isr.umich.edu/data-products/public-survey-data) and the RAND Center for the Study of Aging (https://hrsdata.isr.umich.edu/data-products/rand).

At each wave of HRS, a rotating random 50% subsample of participants were invited to an enhanced interview and given a leave-behind Psychosocial and Lifestyle Questionnaire to return by mail, which included questions on leisure engagement[56]. Participants were eligible to complete this psychosocial questionnaire in 2008 or 2010, which we have combined to form the baseline of our study. In 2008, 8296 participants were invited to complete this questionnaire, and 7073 (85%) returned it. In 2010, 11,213 were eligible, and 8332 (74%) participated. Of the 15,405 participants who participated at baseline, 10,215 also participated in the HRS core survey at our follow-up eight years later (2016/2018) and were thus eligible for inclusion in our study. Of these,

8893 participants had complete data on leisure engagement, and 8771 also participated in the previous wave (in which health behavior covariates were measured), forming our final analytical sample for outcomes measured in the core survey (Table S1). Three additional limitations reduced our sample size further for some outcomes: completion of enhanced physical assessments at follow-up ($n = 7940$), aged 65 and over at follow-up ($n = 4643$), and both restrictions combined ($n = 4131$).

### Leisure engagement

The HRS Social Engagement scale was measured in the psychosocial questionnaire at baseline[56]. This scale included 18 consistent items across 2008 and 2010, three of which were excluded from this study (caring for sick or disabled adults, praying privately, using a computer for email, internet, or other tasks) as they were not typical leisure activities. This left 15 activities, which have previously been categorized into four domains: (a) physical activities (sport/exercise, walking), (b) creative activities (gardening, baking/cooking, needlework, and hobbies), (c) cognitive activities (reading, word games, cards or other games, and writing), and (d) community activities (volunteering, charity work, educational courses, sports or social clubs, non-religious organizations)[28]. Participants reported how frequently they engaged in each activity on a seven-point scale, from never (0) to daily (6). We created an index for each domain, averaging the frequency of engagement in all activities within that domain.

### Experiences of aging

All outcomes were measured at baseline (2008/2010) and eight years later (2016/2018). The subsample in which each outcome was measured is detailed in Table S1.

**Daily functioning.** Included the number of difficulties with ADLs (ranging from 0 to 5; from bathing, eating, dressing, walking across a room, and getting in or out of bed), the number of difficulties with IADLs (ranging from 0 to 5; from using a telephone, taking medication, handling money, shopping, and preparing meals), and number of difficulties with mobility, from walking one block, several blocks, and across a room, jogging one mile, and reaching/extending arms up (0, 1, 2, 3, 4 or more).

**Physical fitness.** Included three self-reported indices, measured as the number of activities with which participants did not have problems within: strength (ranging from 0-3; stooping, kneeling or crouching, pushing or pulling a large object, lifting or carrying weights over ten pounds [like a heavy bag of groceries]), gross motor function (ranging from 0 to 4; walking one block, walking across a room, getting in or out of bed, bathing), and fine motor function (ranging from 0 to 3; picking up a dime, eating, dressing). On each of these outcomes, higher scores indicate better physical fitness. Participants aged 65 and over self-reported whether they had fallen down in the last two years (yes, no; falls). We also included four objective measures of physical fitness, assessed during the enhanced face-to-face HRS interview[57]. Lung function was measured with peak expiratory flow using a Mini-Wright Peak Flow Meter, taken as the average of three measures each 30 s apart. Grip strength was measured with a pistol hand grip device (Smedley spring-type hand dynamometer), taken as the average of two measures on each hand. Static balance was evaluated with three separate, progressively more difficult stances, with a variable derived to indicate completion of these stances (none, side-by-side only, semi-tandem, tandem). Gait speed was measured in participants aged 65 and over with the timed walk test (time to walk a 98.5-in. course twice), with times reversed so that higher scores indicate faster gait speed.

**Heart health.** Was measured with systolic and diastolic blood pressure and pulse, all assessed using an Omron HEM-780 Intellisense

Automated blood pressure monitor with ComFit cuff on the participant's left arm, taken as the average of the final two of three measurements 45–60 s apart. Using the American Heart Association guidelines, we categorized systolic blood pressure as normal (less than 120 mm Hg), elevated (120–129 mm Hg), hypertension stage one (130–139 mm Hg), or hypertension stage two (140 mm Hg and above) and diastolic blood pressure as normal or elevated (less than 80 mm Hg), hypertension stage one (80-89 mm Hg), or hypertension stage two (90 mm Hg and above). Models including these measures were adjusted for whether participants took blood pressure medication (yes, no).

**Weight.** Descriptors were body mass index (BMI), calculated using interviewer-recorded weight and height, and waist circumference, measured with a tape measure at the level of the participant's navel. We categorized BMI using the Centers for Disease Control and Prevention (CDC) guidelines as people who were underweight or healthy weight (below 25), overweight (25 to <30), or people with obesity (30 and above). The underweight and healthy weight categories were combined due to the low proportion of participants who were underweight.

**Sleep.** Measures were self-reported, including whether participants regularly take prescription medication to help them sleep (yes, no) and how often they feel really rested when they wake up in the morning (most of the time, sometimes, rarely/never).

**Long-term physical health problems.** Were self-reported as the number of chronic health conditions (0, 1, 2, 3, 4, 5 or more; from high blood pressure, diabetes, cancer, lung disease, heart disease, stroke, psychiatric problems, arthritis) and the degree of persistent pain experienced (none, mild, moderate, and severe).

**Subjective perceptions of health.** Included participants' ratings of their eyesight using glasses or corrective lenses as needed (excellent, very good, good, fair, poor, and blind), hearing (excellent, very good, good, fair, and poor), and perceived difficulty with balance (never, rarely, sometimes, and often).

### Covariates
Covariates were measured in the HRS core survey at baseline (2008/2010). Demographic factors were age (years), gender (men, women), marital status (married [including cohabiting], unmarried [separated, divorced, widowed, never married]), and race/ethnicity (White [including Caucasian], Black [including African American], Other [including American Indian, Alaskan Native, Asian or Pacific Islander, Hispanic, Other]). Socioeconomic factors were educational attainment (less than high school, high school, college, postgraduate), employment status (employed, retired, not working [including unemployed, temporarily laid off, disabled, homemakers]), pension status (yes, no), total household income (US dollars), total assets (US dollars), and household size (count of other household members). Neighborhood factors were self-reported safety (excellent/good, fair/poor), physical disorder (ranging from 1 to 7; rated presence of vandalism, graffiti, rubbish, vacant, deserted houses, and crime), and social cohesion (ranging from 1 to 7; feels part of this area, trusts people, people are friendly, people will help).

### Statistical analysis
Using an outcome-wide approach[26,27], we tested the associations between frequency of engagement in each leisure domain (physical, creative, cognitive, and community activities) and aging experiences eight years later in regression models. The type of regression was determined by the outcome; negative binomial regression was used for count outcomes (to deal with overdispersion), linear regression for continuous, logistic regression for binary, and ordered logistic regression for ordinal outcomes. Models included all four leisure domains simultaneously and were adjusted for demographic, socioeconomic, and neighborhood covariates and the baseline measure of the outcome. All analyses of heart health were also adjusted for whether participants were taking blood pressure medication (yes, no). The cross-sectional associations between leisure engagement and each experience of aging at baseline are included in the Supplementary Materials, along with the unadjusted models.

We accounted for the complex survey design and attrition using probability weights provided by HRS, including either the weight for the psychosocial questionnaire subsample or the physical measures subsample, dependent on how each outcome was measured. For participants with missing data on aging experiences outcomes or covariates, we imputed data using multiple imputations by chained equations (MICE)[58]. We used Poisson, multinomial logistic, ordered logistic, and logistic regression according to variable type, generating 20 imputed data sets (maximum missing data 16%; Table S2). The imputation model included all variables used in analyses and sampling weights. Separate imputation models were run for each subsample (1) core survey, 2) physical measures, 3) aged 65+, and 4) physical measures and aged 65+; Table S1). All analyses were performed using Stata 17[59].

In sensitivity analyses, we computed *E*-values as indicators of how robust findings were to potential unmeasured confounding[29], using the Stata evalue package[60]. We also performed four additional sensitivity analyses. First, we additionally adjusted analyses for health covariates (cognition, depressive symptoms, prescription medication, psychiatric problems, self-rated health measured at baseline) and health behavior covariates (alcohol use and smoking measured at the wave prior to baseline). Second, as there were concerns about potential bias due to controlling for the outcome at baseline, we also repeated the longitudinal adjusted analyses after omitting the baseline outcome measure. Third, we included different levels of leisure engagement (none, weekly, monthly) to provide a more comprehensive picture of the associations with experiences of aging. Finally, due to concerns around reverse causation, we limited the sample to participants without chronic health conditions at baseline.

### Reporting summary
Further information on research design is available in the Nature Portfolio Reporting Summary linked to this article.

## Data availability
Raw data are available from HRS (https://hrsdata.isr.umich.edu/data-products/public-survey-data) and the RAND Center for the Study of Aging (https://hrsdata.isr.umich.edu/data-products/rand). Derived data supporting the findings of this study are available from the corresponding author, J.K.B., on request. Source data are provided in this paper.

## Code availability
All code for analyses in this study is publicly available online: https://doi.org/10.17605/OSF.IO/8NBXD.

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

## Acknowledgements

We thank Shanae Burch, a thought leader on work at the intersections of the arts, equity, and public health in the US, for her comments on this manuscript. We also gratefully acknowledge the contribution of the HRS study participants. HRS is sponsored by the National Institute on Aging (grant number NIA U01AG009740) and is conducted by the University of Michigan. The EpiArts Lab, a National Endowment for the Arts Research Lab at the University of Florida, is supported in part by an award from the National Endowment for the Arts (1862896-38-C-20; awarded to J.K.S. and D.F.). The opinions expressed are those of the authors and do not represent the views of the National Endowment for the Arts Office of Research & Analysis or the National Endowment for the Arts. The National Endowment for the Arts does not guarantee the accuracy or completeness of the information included in this material and is not responsible for any consequences of its use. The EpiArts Lab is also supported by Americans for the Arts, Bloomberg Philanthropies, the Dharma Endowment Foundation, and the Pabst Steinmetz Foundation (all awarded to J.K.S. and D.F.). D.F. is also supported by the Wellcome Trust (205407/Z/16/Z; D.F.).

## Author contributions

J.K.B., F.B., and D.F. designed the study. J.K.B. conducted the analyses and drafted the paper. All authors (J.K.B., F.B., J.K.S., D.F.) contributed to the writing, made critical revisions, and approved the final paper.

## Competing interests

The authors declare no competing interests.
