## [Peer Review File · Nature Communications]

Reviewers' Comments:

Reviewer #1:

Remarks to the Author:

Using longitudinal data from the Health and Retirement Study (N=8893, 2008/2010 to 2016/2018 questionnaire), the present study examined the association between engagement in leisure activities (physical, creative, cognitive, and community activities were simultaneously included in the models to examine their independent effects) and a total of 23 indicators of aging experiences among older adults. The study suggested a potentially novel and meaningful approach to promoting health in older adults. I have the following suggestions to strengthen the paper.

1) It would be helpful to briefly describe some relevant theories regarding potential mechanisms whereby leisure activities may help reduce aging-related decline. For example, do the four domains of leisure activities share common pathways or do they work through unique pathways to affect health? This would help readers understand the findings.

2) The strongest associations were found with engagement in physical activities. There has been a large body of literature on physical activity and health. It would be helpful to discuss how the findings on physical activities in this study extend the literature. The piece of evidence on creative activities and the null associations with cognitive and community activities are relatively novel, but there are relevant prior studies that could be discussed. For instance, the authors themselves published studies on art engagement and healthy aging using the same dataset, and some of the art activities (e.g., writing, reading, needlework, etc.) were also included as creative and cognitive activities in the present study. There are many prior studies suggesting community participation (e.g., volunteering) is related to better health in older adults. It will be helpful to include more in-depth discussion regarding whether and why the findings in the present study are consistent with or different from prior evidence.

3) There are several potential confounders that may need to be considered. For instance, prior mental illness, cognitive function, and lifestyle factors (e.g., smoking) are all closely related to both the levels of leisure activities (or at least some domains of leisure activities) and to physical health, leading to potential confounding. If I remember it correctly, depression, cognitive function, and lifestyle factors were all measured repeatedly in the Health and Retirement Study. If there are concerns about controlling for potential mediators (e.g., if depression and leisure activities were measured simultaneously at the same wave, controlling for depression could block a potential pathway between leisure activities and subsequent health), the authors may consider adjusting for these potential confounders assessed at prior waves (e.g., the 2006 wave). If adding these additional covariates is not feasible, the authors would need to at least discuss potential confounding by these factors in the paper. The authors used "E-values" to assess potential unmeasured confounding. "E-values" are interpreted considering the covariates that are already adjusted for in the analyses. Therefore, the E-values may be more meaningful after these major confounders are accounted for.

4) Reverse causation (by prior health status) remains a concern. The strong cross-sectional associations that the authors reported actually highlight the importance of controlling for baseline physical health when examining the longitudinal associations between leisure activities and health. If I understand it correctly, in the longitudinal analyses the authors only controlled for the baseline value of a particular health outcome in each model (e.g., when examining BMI as an outcome, the baseline value of BMI was included as a covariate but not any other baseline health indicators were included). Further evidence to reduce concerns about reverse causation would be helpful (e.g., more rigorous control for baseline physical health status, sensitivity analyses excluding participants with major chronic conditions at baseline, etc.).

5) The first sentence in the last paragraph ("We found evidence for persistent positive associations between different domains of leisure engagement ... with a wide range of age-related processes") seems an overstatement of the findings. In this paragraph and several other places in the text, the authors commented that this study suggested the potential of social prescribing to reduce disparities in aging. The present study does not really examine the roles of SES or other disparity related conditions in the association between leisure activities and aging per se. If the author could

cite some prior studies suggesting that leisure activities may benefit health across social strata, it could help make the argument stronger.

Minor comments

6) In Figure 1 and Figure 2 titles, it says "Adjusted coefficients (grey)". Do the authors mean "Adjusted beta coefficients"?

7) In the first paragraph of the Results section, it may be more meaningful to report the standard deviations (e.g., for age, and the frequency of engagement in different activities) rather than the standard errors.

Reviewer #2:

Remarks to the Author:

In this paper, the authors examined the association between four types of leisure activities (physical, cognitive, artistic, and community-engaged) in relation to a comprehensive battery of outcomes related to health problems, many of them associated to aging, in a sample of older adults from the Health and Retirement Study. The main finding was that, among these leisure activities, increased levels of physical activities were associated with less probability of having health problems as well as decreased likelihood of suffering these problems 8 years later. Artistic activities were also found to be associated with lower likelihood of several health problems. This work contributes to current knowledge on several ways. First, by providing evidence of the impact of leisure activities on health in a well characterized population, large enough and with a follow-up that allowed observing changes in the studied outcomes. Second, incorporating to the analyses many components related to common problems associated to unhealthy aging, including physical function and fitness, multimorbidity, sleep problems, or subjective health. Third, by performing appropriate methodology, including adjusting the results by socioeconomic confounders, which have a profound effect on all domains of health.

However, there are two aspects that I am surprised the authors did not consider when doing their study:

1. The inclusion of cognitive function as part of the indicators of unhealthy aging. The authors should justify the lack of information on this outcome. Of minor relevance, it is the short list of chronic diseases considered, or the few variables to define heart health, or the number of current drugs used considered.
2. The lack of information about other lifestyles in the analyses. Diet, alcohol and tobacco consumption, and sedentariness have a notorious impact on health. The authors cannot avoid this when quantifying the impact of leisure activities on health. Since these variables have been measured in the HRS, I would expect they were considered potential confounders and included in the adjusted models. This merits an explanation by the authors. The E-value calculation does not seem to solve this issue.
3. The models have included as the exposure an increase in leisure time in relation to each outcome. Although a dose-response association is always preferable to suggest causality in the associations, assessing the association at several levels of the exposure always provide a more comprehensive picture. Since the authors are studying many outcomes, these analyses might be overwhelming, but many readers would be interested in knowing whether the decreased risk associated with some of the leisure activities can be reached at low levels of exposure or higher levels are required to obtain benefit.

Esther Lopez-Garcia, PhD
Universidad Autónoma de Madrid, Spain

Reviewer #3:

Remarks to the Author:

This study explored the cross-sectional and longitudinal associations between different dimensions of leisure engagement with different aspects of aging (daily functioning, physical fitness, long-term physical health problems, heart health, weight, sleep, subjective perceptions of health) using data

on 8,893 older adults from the U.S. Health and Retirement Study.

Focusing on the longitudinal results only (for which causal inference is more strongly supported), the main findings center around leisure-time physical activity, whereas for creative and cognitive activities, the associations are either modest or null, and for social domains ("community activities"), the associations are either null or in the opposite direction to expected.

As the authors point out, the health benefits of physical activity are already well-established in the literature. Hence, this does not make any unique contributions to the literature. For other domains of leisure activity, the lack of associations predominates and do not appear to make a convincing contribution either that would deem them noteworthy that would support the hypothesized associations.

Methodologically and from an interpretation perspective, I have questions and/or concerns about the following issues:

1. The Health and Retirement Study is a longitudinal cohort with follow up starting as early as 1992. How did the authors address the potential for selection bias due to attrition by the time of baseline in 2008/2010 in this cohort?
2. There is the potential bias from controlling for the outcome variable with the same variable at baseline. How did the results change when either the baseline variable was omitted, or did the authors consider the outcome of change in the outcome from baseline i.e., with the outcome variable corresponding to change from baseline?
3. It was not clear to me whether the leisure domains were co-adjusted in all of the models, since they could conceivably lead to confounding if omitted. Which approach did the authors take? If co-adjusted, was there any evidence of collinearity?
4. There is the potential for residual confounding by other omitted factors. What factors that were not controlled for could have led to residual confounding that could explain the associations or lack of associations, given the e-values estimated?

Reviewer #1

Using longitudinal data from the Health and Retirement Study (N=8893, 2008/2010 to 2016/2018 questionnaire), the present study examined the association between engagement in leisure activities (physical, creative, cognitive, and community activities were simultaneously included in the models to examine their independent effects) and a total of 23 indicators of aging experiences among older adults. The study suggested a potentially novel and meaningful approach to promoting health in older adults. I have the following suggestions to strengthen the paper.

1) It would be helpful to briefly describe some relevant theories regarding potential mechanisms whereby leisure activities may help reduce aging-related decline. For example, do the four domains of leisure activities share common pathways or do they work through unique pathways to affect health? This would help readers understand the findings.

Reply: Thank you for this suggestion. We have now described a theoretical model linking leisure engagement to health in the introduction, including potential pathways through which leisure activities might influence age-related decline, and how activities should be grouped (page 3 lines 14-19). Additionally, in the discussion, we have noted that physical activity may affect health through unique pathways, not activated by other domains of leisure activities (page 6 lines 32-35). We also discuss the mechanisms through which creative and community activities may prevent decline (page 6 lines 14-17).

2) The strongest associations were found with engagement in physical activities. There has been a large body of literature on physical activity and health. It would be helpful to discuss how the findings on physical activities in this study extend the literature. The piece of evidence on creative activities and the null associations with cognitive and community activities are relatively novel, but there are relevant prior studies that could be discussed. For instance, the authors themselves published studies on art engagement and healthy aging using the same dataset, and some of the art activities (e.g., writing, reading, needlework, etc.) were also included as creative and cognitive activities in the present study. There are many prior studies suggesting community participation (e.g., volunteering) is related to better health in older adults. It will be helpful to include more in-depth discussion regarding whether and why the findings in the present study are consistent with or different from prior evidence.

Reply: As recommended, we have outlined in the discussion that our findings extend previous evidence by showing the broad positive impacts of physical activity across domains, allowing for comparison across health outcomes, demonstrating that these associations are independent of participation in creative, community, and cognitive activities (page 6 lines 28-32). We have also added further discussion of previous literature examining creative, cognitive, and community engagement and older adults' health, discussing how our findings compare to previous evidence (page 7 lines 5-27).

3) There are several potential confounders that may need to be considered. For instance, prior mental illness, cognitive function, and lifestyle factors (e.g., smoking) are all closely related to both the levels of leisure activities (or at least some domains of leisure activities) and to physical health,

leading to potential confounding. If I remember it correctly, depression, cognitive function, and lifestyle factors were all measured repeatedly in the Health and Retirement Study. If there are concerns about controlling for potential mediators (e.g., if depression and leisure activities were measured simultaneously at the same wave, controlling for depression could block a potential pathway between leisure activities and subsequent health), the authors may consider adjusting for these potential confounders assessed at prior waves (e.g., the 2006 wave). If adding these additional covariates is not feasible, the authors would need to at least discuss potential confounding by these factors in the paper. The authors used “E-values” to assess potential unmeasured confounding. “E-values” are interpreted considering the covariates that are already adjusted for in the analyses. Therefore, the E-values may be more meaningful after these major confounders are accounted for.

Reply: After considering the full set of relevant measured confounders, we have additionally adjusted the main analyses for further socioeconomic and neighborhood factors (pension status, total assets, neighborhood physical disorder, neighborhood social cohesion). As suggested, we have also performed a sensitivity analysis adjusted for health covariates (cognition, depressive symptoms, prescription medication, psychiatric problems, self-rated health) and health behaviors (alcohol use, smoking). As the reviewer suggested, we attempted to adjust for all health and health behavior covariates at the wave preceding baseline (2006/2008). However, due to the extent of missing data in health covariates at this wave, and issues with convergence and imputation of missing values during imputation, we were only able to include health behaviors measured prior to baseline. Health covariates are measured at baseline. Due to the noted concerns around health-related factors being on the causal pathway between leisure engagement and experiences of aging, we have included this as a sensitivity analysis. Both health factors and health behaviors are likely to be relatively stable in older adults meaning that, regardless of whether they were measured in the wave preceding baseline or at baseline, they are still likely to represent potential mediators.

We have described the new covariates in detail in the Supplementary Materials and added a fully adjusted sensitivity analysis (Table S6) with updated results and e-values. We have discussed these analyses and also noted that unmeasured confounding may still be an issue due to factors such as urbanicity and social support in the strengths and limitations section of the discussion (page 8 lines 23-25).

4) Reverse causation (by prior health status) remains a concern. The strong cross-sectional associations that the authors reported actually highlight the importance of controlling for baseline physical health when examining the longitudinal associations between leisure activities and health. If I understand it correctly, in the longitudinal analyses the authors only controlled for the baseline value of a particular health outcome in each model (e.g., when examining BMI as an outcome, the baseline value of BMI was included as a covariate but not any other baseline health indicators were included). Further evidence to reduce concerns about reverse causation would be helpful (e.g., more rigorous control for baseline physical health status, sensitivity analyses excluding participants with major chronic conditions at baseline, etc.).

Reply: Thank you for these suggestions. We hope that the addition of health covariates, as described above, should ensure much more rigorous control for baseline health status. We have

added further sensitivity analyses as advised limiting the sample to participants without chronic health conditions at baseline (page 6 lines 5-9, Table S9). This resulted in small sample sizes (n=395 to n=1460), meaning there was very little evidence for associations between leisure engagement and experiences of aging. However, coefficients and confidence intervals were generally in line with the main analyses.

Please note that, given the number of sensitivity analyses recommended by all reviewers, we have moved the concurrent analyses to the supplement to ensure the results section is not too long. However, we have still summarized the cross-sectional findings in the results (page 5 lines 12-19) and described issues of reverse causality in the discussion (page 8 lines 12-14).

5) The first sentence in the last paragraph (“We found evidence for persistent positive associations between different domains of leisure engagement ... with a wide range of age-related processes”) seems an overstatement of the findings. In this paragraph and several other places in the text, the authors commented that this study suggested the potential of social prescribing to reduce disparities in aging. The present study does not really examine the roles of SES or other disparity related conditions in the association between leisure activities and aging per se. If the author could cite some prior studies suggesting that leisure activities may benefit health across social strata, it could help make the argument stronger.

Reply: We have made sure not to overstate how findings, toning down our language. We have also added supporting evidence that leisure engagement can support mental and physical health across social strata, potentially with stronger effects in more deprived areas (page 8 lines 43-44).

Minor comments

6) In Figure 1 and Figure 2 titles, it says “Adjusted coefficients (grey)”. Do the authors mean “Adjusted beta coefficients”?

Reply: Thank you for spotting this. We did mean beta coefficients, so have updated the figure titles accordingly.

7) In the first paragraph of the Results section, it may be more meaningful to report the standard deviations (e.g., for age, and the frequency of engagement in different activities) rather than the standard errors.

Reply: We have changed all of the standard errors to standard deviations as suggested (page 4 lines 4-10).

Reviewer #2

In this paper, the authors examined the association between four types of leisure activities (physical, cognitive, artistic, and community-engaged) in relation to a comprehensive battery of outcomes related to health problems, many of them associated to aging, in a sample of older adults from the

Health and Retirement Study. The main finding was that, among these leisure activities, increased levels of physical activities were associated with less probability of having health problems as well as decreased likelihood of suffering these problems 8 years later. Artistic activities were also found to be associated with lower likelihood of several health problems.

This work contributes to current knowledge on several ways. First, by providing evidence of the impact of leisure activities on health in a well characterized population, large enough and with a follow-up that allowed observing changes in the studied outcomes. Second, incorporating to the analyses many components related to common problems associated to unhealthy aging, including physical function and fitness, multimorbidity, sleep problems, or subjective health. Third, by performing appropriate methodology, including adjusting the results by socioeconomic confounders, which have a profound effect on all domains of health.

However, there are two aspects that I am surprised the authors did not consider when doing their study:

1. The inclusion of cognitive function as part of the indicators of unhealthy aging. The authors should justify the lack of information on this outcome. Of minor relevance, it is the short list of chronic diseases considered, or the few variables to define heart health, or the number of current drugs used considered.

Reply: Thank you for your comments. In this paper, we have focused on the experiences of aging related to physical health, as opposed to cognition, mental health, wellbeing, or any other outcomes. There is extensive existing evidence of the benefits of leisure engagement for these outcomes, but less research has compared the associations between different types of leisure engagement and a wide range of physical health outcomes. In terms of chronic diseases, we have included conditions that are repeatedly measured across waves in HRS, as indexed by the generated variable in the HRS RAND longitudinal dataset. We believe that this captures the key most prevalent chronic conditions, which are of most relevance to older adults. Similarly, for heart health, we have included variables objectively measured in the HRS nurse visit. Self-reported heart disease is also included in the chronic disease variable. Based on the recommendations of the other reviewers, we have now performed a sensitivity analysis that is adjusted for health factors, including cognition and number of disorders for which prescription medications are taken (from cholesterol, pain in joints/muscles, allergies/asthma/breathing, stomach problems, sleep, anxiety/depression) at baseline (page 5 lines 29-34, Table S6).

2. The lack of information about other lifestyles in the analyses. Diet, alcohol and tobacco consumption, and sedentariness have a notorious impact on health. The authors cannot avoid this when quantifying the impact of leisure activities on health. Since these variables have been measured in the HRS, I would expect they were considered potential confounders and included in the adjusted models. This merits an explanation by the authors. The E-value calculation does not seem to solve this issue.

Reply: Given these suggestions, alongside the feedback from reviewer 1, we have performed a sensitivity analysis adjusted for health covariates (cognition, depressive symptoms, prescription

medication, psychiatric problems, self-rated health) and health behaviors (alcohol use, smoking). Data on diet and sedentariness (distinct from physical activity) were not available in the included waves of HRS. Due to concerns around health-related factors being on the causal pathway between leisure engagement and experiences of aging, we attempted to adjust for all health and health behavior covariates at the wave preceding baseline (2006/2008). However, because of the extent of missing data in health covariates at this wave, and issues with convergence and imputation of missing values during imputation, we were only able to include health behaviors measured prior to baseline. Health covariates are measured at baseline. Both health factors and health behaviors are likely to be relatively stable in older adults meaning that, regardless of whether they were measured in the wave preceding baseline or at baseline, they are still likely to represent potential mediators. We have thus included this as a sensitivity analysis (page 5 lines 29-34, Table S6).

3. The models have included as the exposure an increase in leisure time in relation to each outcome. Although a dose-response association is always preferable to suggest causality in the associations, assessing the association at several levels of the exposure always provide a more comprehensive picture. Since the authors are studying many outcomes, these analyses might be overwhelming, but many readers would be interested in knowing whether the decreased risk associated with some of the leisure activities can be reached at low levels of exposure or higher levels are required to obtain benefit.

Reply: Thank you for this interesting recommendation. We have added a sensitivity analysis including different levels of leisure engagement to provide a more comprehensive picture of the associations with experiences of aging (Tables S8A-S8B). We categorized engagement frequencies into none, monthly, and weekly engagement. This indicated that, for physical activities, most of the associations with experiences of aging were driven by more frequent weekly engagement. Participants who only engaged monthly experienced fewer benefits. In contrast, for creative activities, findings were more similar across monthly and weekly engagement frequencies, showing less of a dose-response relationship (page 5 line 41 – page 6 line 3).

Esther Lopez-Garcia, PhD
Universidad Autónoma de Madrid, Spain

Reviewer #3

This study explored the cross-sectional and longitudinal associations between different dimensions of leisure engagement with different aspects of aging (daily functioning, physical fitness, long-term physical health problems, heart health, weight, sleep, subjective perceptions of health) using data on 8,893 older adults from the U.S. Health and Retirement Study.

Focusing on the longitudinal results only (for which causal inference is more strongly supported), the main findings center around leisure-time physical activity, whereas for creative and cognitive activities, the associations are either modest or null, and for social domains ("community activities"), the associations are either null or in the opposite direction to expected.

As the authors point out, the health benefits of physical activity are already well-established in the literature. Hence, this does not make any unique contributions to the literature. For other domains of leisure activity, the lack of associations predominates and do not appear to make a convincing contribution either that would deem them noteworthy that would support the hypothesized associations.

Methodologically and from an interpretation perspective, I have questions and/or concerns about the following issues:

1. The Health and Retirement Study is a longitudinal cohort with follow up starting as early as 1992. How did the authors address the potential for selection bias due to attrition by the time of baseline in 2008/2010 in this cohort?

Reply: HRS employs a steady-state design, replenishing the sample every six years with younger cohorts, including cohorts added in 2004 and 2010. As this does not address attrition in the oldest participants, we used probability weights provided by HRS to account for non-response, both to the HRS core survey and to the psychosocial and lifestyle questionnaire. HRS calculated the probability of non-response according to a range of factors, including age, sex, race/ethnicity, coupleness, education, work status, self-rated health, counts of functional limitations (Nagi, IADL, ADL), vision rating, cognitive status, and religious attendance. We have noted that these weights also account for attrition in the methods (page 11 line 13).

2. There is the potential bias from controlling for the outcome variable with the same variable at baseline. How did the results change when either the baseline variable was omitted, or did the authors consider the outcome of change in the outcome from baseline i.e., with the outcome variable corresponding to change from baseline?

Reply: Thank you for this suggestion. We have added a sensitivity analysis omitting the outcome variable measured at baseline. Evidence for associations between leisure activities and experiences of aging remained similar, albeit slightly stronger for physical, creative, and cognitive activities (page 5 lines 36-39, Table S7). We believe that this is a better alternative than using a change score as the outcome, which do not provide meaningful causal-effect estimates and may obscure estimands, potentially diverging substantially in magnitude and direction from the causal effects (Tennant, Arnold, Ellison & Gilthorpe, 2022).

**Tennant, Arnold, Ellison, Gilthorpe, (2022) Analyses of 'change scores' do not estimate causal effects in observational data. *International Journal of Epidemiology*, 51 (5), 1604–1615.
<https://doi.org/10.1093/ije/dyab050>**

3. It was not clear to me whether the leisure domains were co-adjusted in all of the models, since they could conceivably lead to confounding if omitted. Which approach did the authors take? If co-adjusted, was there any evidence of collinearity?

Reply: The leisure domains were co-adjusted in all of the models, but there was no evidence of collinearity. Correlations between leisure domains ranged from $r=0.21$ to $r=0.34$. We have now made this co-adjustment clearer in the statistical analyses (page 11 lines 6-7), and reported the correlations in the results (page 4 lines 10-11).

4. There is the potential for residual confounding by other omitted factors. What factors that were not controlled for could have led to residual confounding that could explain the associations or lack of associations, given the e-values estimated?

Reply: As also suggested by the other reviewers, we have now adjusted all analyses for additional socioeconomic factors (pension status, total assets, neighborhood physical disorder, neighborhood social cohesion). We have additionally performed a sensitivity analysis for health covariates (cognition, depressive symptoms, prescription medication, psychiatric problems, self-rated health), and health behaviors (alcohol use, smoking), as these may be on the causal pathway (page 5 lines 29-34). We have also noted that unmeasured confounding may still be an issue due to factors such as urbanicity, diet, and social support in limitations section of the discussion (page 8 lines 23-28).

Reviewers' Comments:

Reviewer #1:

Remarks to the Author:

The authors have sufficiently addressed my comments, and I have no further major comments.

Minor points:

- The experiences of aging could involve multiple domains. It would be helpful to add a sentence in the manuscript clarifying that this study focuses on the aspect of physical health in aging.

- On page 8 line 10, it seems the references are incorrectly inserted in the sentence "This study has a number of strengths".

Reviewer #2:

None

Reviewer #3:

Remarks to the Author:

The authors have responded adequately to my concerns raised.

We thank the reviewers for reconsidering our paper, and for their positive feedback. There were just two minor comments that we respond to below.

Reviewer #1 (Remarks to the Author):

The authors have sufficiently addressed my comments, and I have no further major comments.

Minor points:

- The experiences of aging could involve multiple domains. It would be helpful to add a sentence in the manuscript clarifying that this study focuses on the aspect of physical health in aging.

Reply: We have added the following to the introduction “In this study, we take an outcome-wide approach, assessing experiences of aging related to physical health.”

- On page 8 line 10, it seems the references are incorrectly inserted in the sentence “This study has a number of strengths”.

Reply: Thank you for spotting this. We have now removed these references, which were inserted in error.

Reviewer #3 (Remarks to the Author):

The authors have responded adequately to my concerns raised.

Reply: Thank you for reviewing our manuscript.